# Tree-ensemble analysis assesses presence of multifurcations in single cell data

Will Macnair iD, Laura De Vargas Roditi[†], Stefan Ganscha & Manfred Claassen[*] iD

## Abstract

We introduce TreeTop, an algorithm for single cell data analysis to identify and assign a branching score to branch points in biological processes which may have multi-level branching hierarchies. We demonstrate branch point identification for processes with varying topologies, including T-cell maturation, B-cell differentiation and hematopoiesis. Our analyses are consistent with recent experimental studies suggesting a shallower hierarchy of differentiation events in hematopoiesis, rather than the classical multi-level hierarchy.

**Keywords** computational biology; proteomics; single cell RNA-seq; trajectory inference

**Subject Categories** Computational Biology; Genome-Scale & Integrative Biology; Methods & Resources

**Mol Syst Biol. (2019) 15: e8552**

## Introduction

Many important biological processes, such as differentiation in developmental and immune biology, and clonal evolution in cancer, can be conceived of as bi- or multi-furcated cellular state trajectories. Hematopoiesis is such a process, where hematopoietic stem cells (HSCs) give rise to multiple distinct mature blood cell types via a sequence of lineage commitments. The exact sequence is still debated (Perié & Duffy, 2016), either assuming a hierarchical architecture of multiple fate decisions via distinct oligopotent progenitor cell states (Weissman *et al*, 2001; Orkin & Zon, 2008; Seita & Weissman, 2010), or a simpler hierarchy of hematopoiesis, with very few oligopotent progenitors, where multipotent cells differentiate directly into committed lineages (Paul *et al*, 2015; Notta *et al*, 2016; Velten *et al*, 2017).

High-dimensional single cell technologies, such as single cell RNA sequencing (Tang *et al*, 2009) and mass cytometry (Bendall *et al*, 2014), constitute widely used tools to investigate such opposing models of differentiation, and other branching processes. These technologies allow the evaluation of the state of single cells, i.e. the transcriptional or proteomic abundance profile in the case of single cell RNA sequencing or mass cytometry, respectively. Biological processes can be conceived of as trajectories through state space: ordered sequences of cellular states that can either be derived from time series or reconstructed from non-time series single cell data (Stegle *et al*, 2015). We define a *branch point* as the location in state space where three or more distinct cellular state trajectories meet. Branch points dissect these trajectories into distinct state trajectory branches.

Identifying branch points is challenging because for each single cell measurement, both branch membership and ordering within each branch must be learned simultaneously. SPADE was the first approach to fitting multiple branches, by fitting a single minimum spanning tree to non-deterministically clustered data (Bendall *et al*, 2011). Monocle fits smoothed trees to a low-dimensional representation of single cell data, where branch points in the tree are assumed to correspond to branch points in the data (Trapnell *et al*, 2014; Qiu *et al*, 2017). Both Monocle and SPADE by definition impose a tree topology, regardless of the actual topology of the data. Wishbone (Setty *et al*, 2016) and diffusion pseudotime (Haghverdi *et al*, 2016) both use an embedding whose distances correspond to those along the underlying low-dimensional manifold, representing the data via diffusion maps (Coifman *et al*, 2005). Distinct branches are then identified via anti-correlations in graph distances to a selected root point that has to be sensibly defined *a priori*. SCUBA uses bifurcation theory of dynamical systems to determine the presence or absence of branching points, but requires the data to have annotations of time (Marco *et al*, 2014). TSCAN first clusters the data, then like Monocle infers a tree from the data (Ji & Ji, 2016). p-Creode fits multiple trees to the data, uses the data to smooth them, then identifies the tree which is most central within these (Herring *et al*, 2017). These algorithms can all (with the exception of SCUBA, which requires time annotations) return branch points regardless of the actual evidence in the data, and any decision on the presence or absence of a branch point must be made by the user.

## Results and Discussion

We introduce TreeTop to address these shortcomings: in particular, the inability of other algorithms to inform users whether or not a branch point is present, and in addition the supervised root point

---

Institute of Molecular Systems Biology, ETH Zurich, Zurich, Switzerland
*Corresponding author. Tel: +41 44 633 07 33; E-mail: claassen@imsb.biol.ethz.ch
†Present address: Institute of Pathology and Molecular Pathology, University Hospital Zurich, Zurich, Switzerland

selection of Wishbone, and the strong topological assumptions of Monocle. TreeTop takes as input high-dimensional single cell measurements. Via a representation of the data as an ensemble of trees, it identifies branch points which may join more than three branches, and assigns a relative branching score to the identified branch points. TreeTop first approximates the topology of the input dataset by an ensemble of trees (Fig 1A). A set of reference nodes, representing subpopulations of cells with similar states, is selected by the algorithm. These are connected by sampled trees, each of

which may have different edges connecting the nodes, capturing the spectrum of possible transitions between states of the underlying biological process. Secondly, each node is scored for branching by quantifying how consistently cutting each tree at that node partitions the ensemble of trees into separate branches (Fig 1B, Materials and Methods).

To assess how confident we are in a branch point derived from a dataset, we normalize this score, by comparison to reference score distributions: distributions of scores calculated from synthetic data

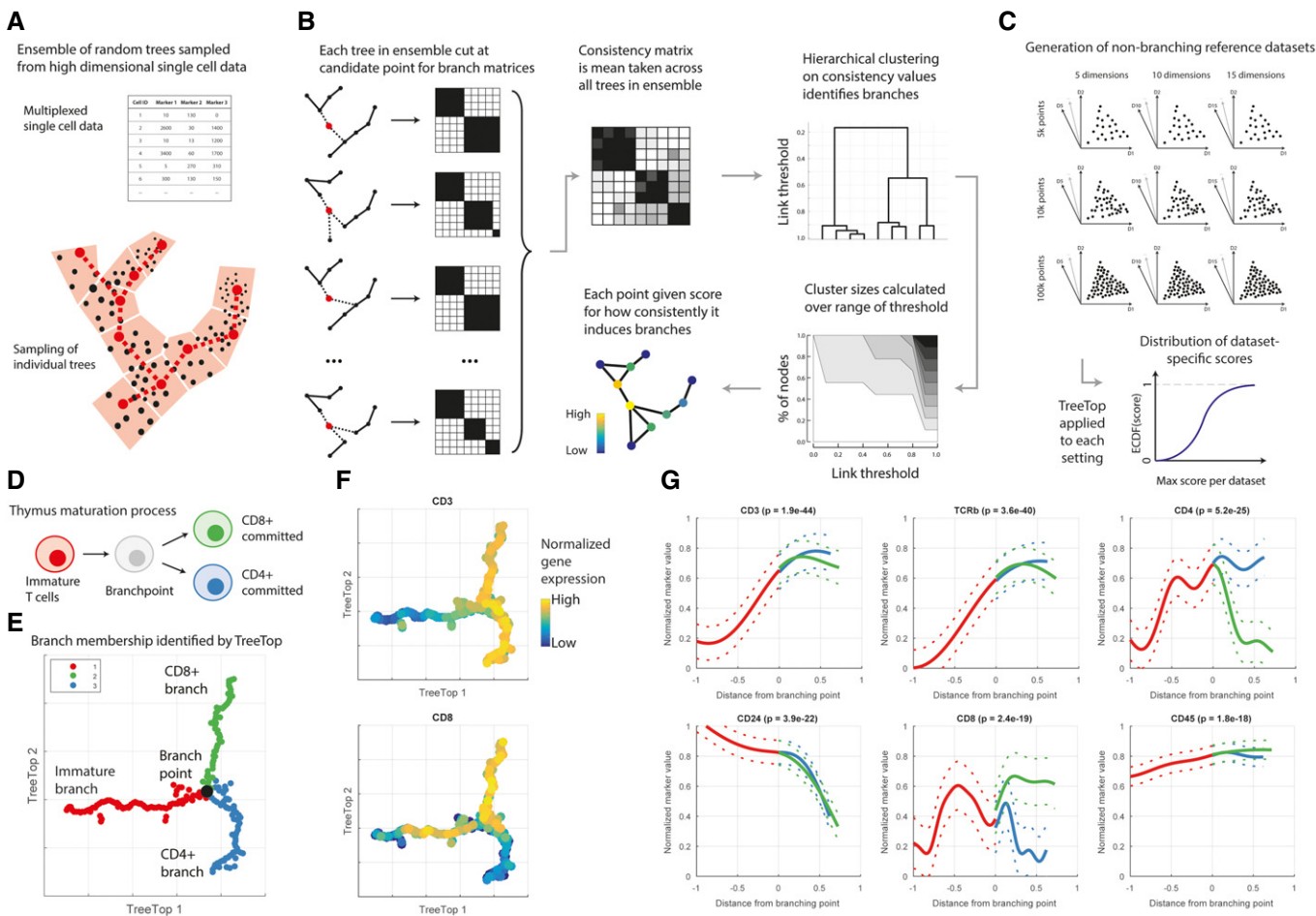

**Figure 1.  TreeTop methodology and demonstration.**

A   Reference nodes are selected to be evenly distributed through data via seeding algorithm for k-means (Moseley *et al*, 2012); all other cells are assigned to closest reference node. Each tree is sampled by selecting a cell from each partition uniformly at random, then joining these points via a minimum spanning tree.

B   To assess each reference node as a branch point (red), each tree in the ensemble is cut (removed edges are shown dashed), partitioning the remaining cells. Mean pairwise co-occurrence across all branch matrices stored in consistency matrix, i.e. *(i, j)* entry is proportion of trees in which cells *i* and *j* were in the same branch, when cut at the red cell. Hierarchical clustering (single linkage) is performed on each consistency matrix. Sizes of the largest clusters are then calculated over all possible dendrogram cut heights, and used to score each point for branching; raw branching score is mean size of third and smaller clusters over all thresholds.

C   Cartoon of reference datasets based on randomly generated synthetic datasets defined to contain structure but no branch points, over a range of parameters for comparison with different sizes of input datasets.

D   Cartoon of cell types in maturation of T cells in thymus.

E   Force-directed graph layout of mass cytometry thymus data [30 antibodies used (Setty *et al*, 2016)], pre-processed with diffusion maps (Materials and Methods; Coifman *et al*, 2005). Point with highest relative branching score (black) is reported branch point, although more than one point may have a score indicating branching. Colours indicate identified branches.

F   TreeTop layout annotated with abundances of selected proteins.

G   Abundance profiles for proteins on branches; *x*-axis is mean tree distance from identified branch point. Proteins selected for most significant differences between branches. Significance is calculated via ANOVA applied to marker abundances for each branch, Bonferroni-corrected.

defined to not contain branch points, but to be comparable to the considered experimental data, i.e. having the same number of observations and dimensionality as the test data. This procedure results in a *relative branching score* indicating how much more branching was observed, relative to that observed in simple non-branching topologies. This allows us to state whether there is evidence that a given input dataset supports branching (Fig 1C, Materials and Methods). Our method therefore suggests where branching processes are likely, to be subsequently confirmed by further experimental work.

Alternative methods for identifying branch points always return the optimal branch point identified, regardless of whether one is supported by the data. We applied TreeTop, Wishbone and Monocle to non-branching synthetic datasets, including those used to define reference score distributions for data from non-branching processes (Appendix Fig S1A–D). Both Wishbone and Monocle identified three non-trivial branches in such datasets, where none is present. TreeTop's relative branching score is defined with reference to datasets such as these, specifically to avoid false-positive branch point identifications.

Assessment of the presence of branch points has typically been done qualitatively, via visual inspection of suitable projections. However, we have found these to be misleading: the embeddings found by Monocle identify exclusively trees, regardless of the topology of the data (Appendix Fig S1B). t-SNE projections (van der Maaten & Hinton, 2008), as used by Wishbone, are

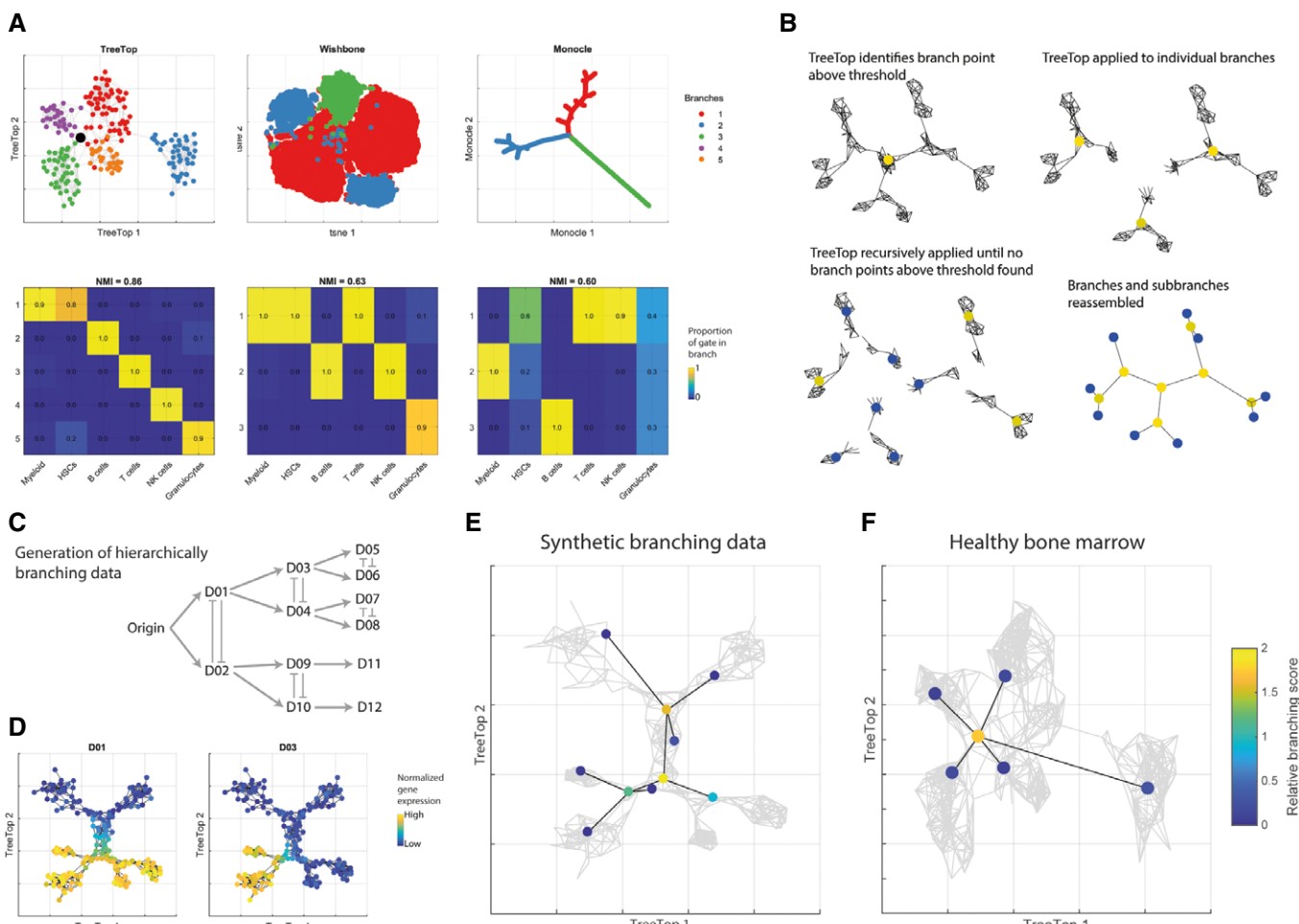

**Figure 2.  TreeTop applied to mass cytometry data is consistent with shallow hierarchy model of hematopoiesis.**

A    TreeTop, Wishbone and Monocle compared on healthy human bone marrow mass cytometry data (Amir *et al*, 2013). TreeTop and Wishbone applied to full dataset, Monocle applied to sample of 2,000 cells. Top row shows layouts and branches identified by each method. Bottom row shows specificity of manual gate allocation to identified branches: heading of plot shows normalized mutual information (NMI; Danon *et al*, 2005) across all gates, branches. Ungated populations excluded. Plots showing layouts annotated by markers used are shown in Appendix Fig S8.

B    Cartoon of recursive application of TreeTop to identify hierarchies of branch points. TreeTop decides whether to recurse at each step by reference to relative branching score.

C    Cartoon of generation of hierarchically branching synthetic data, following classical hematopoietic architecture (Weissman *et al*, 2001; Materials and Methods).

D    TreeTop layout of synthetic deep hierarchy branching data as positive control, annotated species selected to illustrate branch evolution.

E    Result of recursive application of TreeTop to synthetic branching data, showing identification of hierarchy of branch points.

F    Result of recursive application of TreeTop to healthy human bone marrow data, showing only one branch point identified.

                                                     

independent of the branch inference procedure, and frequently do not respect the continuity of the underlying process (Appendix Fig S1C). By representing the data by an *ensemble* of trees, rather than one individual tree, TreeTop is able to represent a much wider class of underlying topologies, including those with cycles (representing a cycle with one individual tree requires the tree to be cut at some point; with an ensemble of trees, each tree can be cut at a different point, allowing the full cycle to be represented). TreeTop visualizes the learned ensemble of trees via a force-based graph layout and provides a flexible and interpretable layout for the input dataset: comparison to the first two principal components of sample synthetic datasets shows that TreeTop's graph-based visualization accurately captures the global structure of the data (Appendix Fig S1D).

We assessed TreeTop's capability to suggest the presence or absence of branch points for biological processes with different known topologies. We applied TreeTop to mass cytometry data of T cells undergoing maturation in the thymus (Setty *et al*, 2016), a

process known to comprise a simple branch point with well-understood state transitions (Fig 1D). TreeTop assigns a score of 1.8 for this branch point (where 1 is the maximum score observed in comparison non-branching topologies). The layout shows three clearly distinct branches (Fig 1E), and the markers identified as showing the greatest difference between branches are consistent with biological expectations (Fig 1F and G, Appendix Fig S2). Application of TreeTop to B-cell maturation mass cytometry data (Bendall *et al*, 2014), a linear process, gave a score of 0.97, indicating no branching, or weak evidence of branching, consistent with the expected consecutive changes in marker abundances. TreeTop's layout also shows a linear structure (Appendix Fig S3). Performance of the alternative methods above was mixed, with Wishbone identifying large branches in the linear B-cell maturation data (Appendix Figs S4 and S5), and Monocle identifying a cluster of smaller spurious branches (Appendix Figs S6 and S7). The flexibility of the ensemble of trees learned by TreeTop permits the accurate assessment of a wide range of topologies.

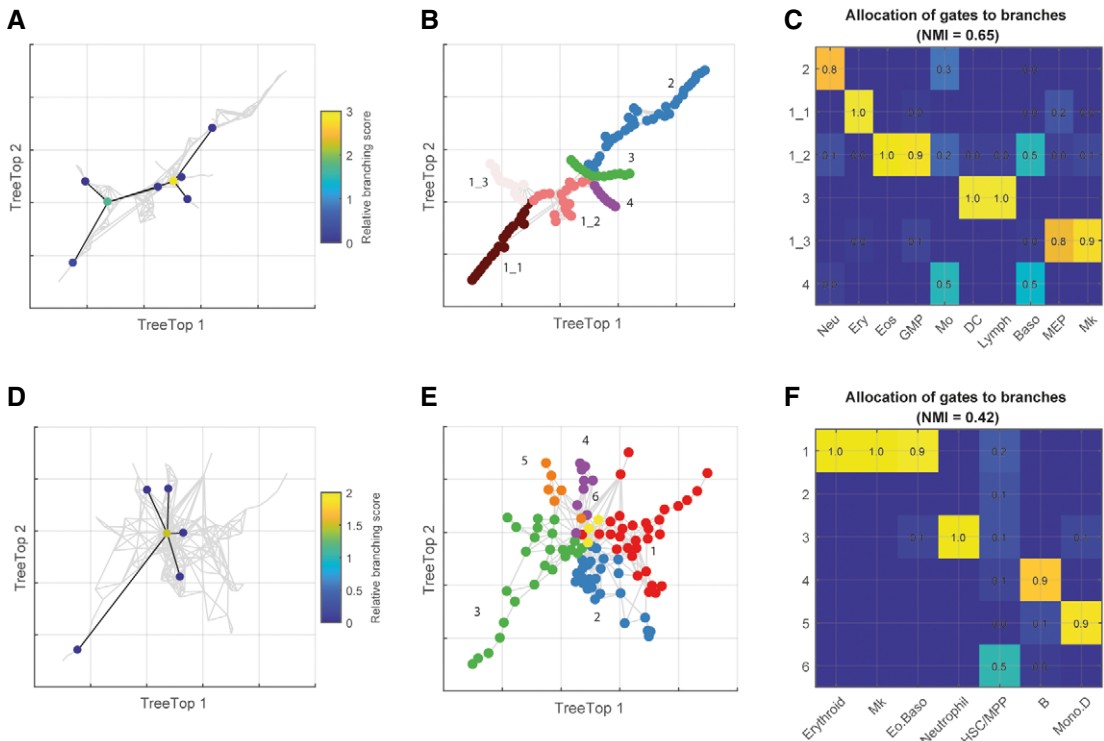

**Figure 3.  TreeTop applied to single cell RNA-seq is consistent with an erythroid-myeloid branch point.**

A–C   TreeTop applied to single cell RNA-seq data derived from 2,730 haematopoietic stem cells taken from healthy human bone marrow (Paul *et al*, 2015). (A) Results of recursive application of TreeTop to data from Paul *et al*, which finds a branch point separating Erythroid, Megakaryocyte and other cell types, then separates additional cell types. TreeTop applied to first eight diffusion components, based on "elbow" criterion. (B) Colours correspond to branches identified by TreeTop, displayed over TreeTop layout. Underscores denote sub-branches; i.e., 1_1, . . ., 1_4 are the sub-branches of branch 1 identified by TreeTop. (C) Contingency table showing matching between branches identified by TreeTop and labels from Paul *et al* (Ery, Erythroid; Eos, Eosinophil; GMP, Granulocyte–macrophage progenitor; MEP, Megakaryocyte–erythroid progenitor; Baso, Basophil; Mo, Monocyte; Neu, Neutrophil; Mk, Megakaryocyte; DC, Dendritic Cell; Lymph, Lymphoid). Labels not used in TreeTop calculations.

D–F   TreeTop applied to single cell RNA-seq data derived from 1,034 haematopoietic stem and pluripotent cells taken from healthy human bone marrow (Velten *et al*, 2017). Data from donor 1 used, pre-processed (variance stabilizing transform) as in original paper. TreeTop applied to diffusion map processed data. (D) Results of recursive application of TreeTop to data from Velten *et al*, which finds one branch point. TreeTop applied to first 11 diffusion components, based on "elbow" criterion. (E) Branches identified by TreeTop, displayed over TreeTop layout. (F) Contingency table showing matching between branches identified by TreeTop and labels from Velten *et al* paper. Labels not used in TreeTop calculations. Labels taken from STEMNET classifier (Velten *et al*, 2017) applied to processed mRNA species abundance values (Mk, Megakaryocyte; Eo.Baso, Eosinophil/Basophil; B, B cell; Mono-D, Monocyte/Dendritic Cell).

TreeTop enables us to assess whether branch points may be present in datasets whose structure is uncertain. We examined the two competing models for hematopoiesis, namely a deep or a shallower hierarchy, by applying TreeTop to mass cytometry data of healthy human bone marrow (Amir *et al*, 2013). TreeTop suggests a branch point (score = 1.7) connecting five distinct branches (Fig 2A, Appendix Fig S8); the ensemble of trees allows branch points connecting more than three branches to be identified. Inspection of marker expression within the branches suggests that these correspond to T cells (CD3$^+$), NK cells (CD7$^+$), myeloid cells (CD33$^+$), B cells (CD19$^+$ CD20$^+$), and granulocytes (CD24$^+$/CD15$^+$; gating description in Appendix Fig S9). These branches are connected by a branch point that does not express any markers for differentiated cell types, corresponding to the HSC compartment, consistent with its expected central position within the hematopoietic process. TreeTop enables us, in contrast to currently available approaches, to assess the presence of further subsequent branch points supporting the deep hierarchy model (Fig 2B). To confirm that TreeTop is able to find hierarchies of branch points, we applied TreeTop recursively to synthetic data specified to be representative of mass cytometry data and to include a hierarchy of branch points (Fig 2C–E, Appendix Figs S10 and S11, Materials and Methods), and identified clear further branches. We examined the presence of further branch points in the bone marrow data by applying TreeTop recursively to each of the five identified branches; it did not suggest further branch points within the branches (Fig 2F). This result is consistent with no further, deeper branch points within the set of markers measured.

In addition to analysing mass cytometry data, we also applied TreeTop to two single cell RNA-seq datasets from healthy human bone marrow (Fig 3, Materials and Methods). Applied to the data obtained by Paul *et al* (2015), TreeTop identified first a branch point separating Erythroid/MEP, Megakaryocyte and the remaining cell types (score = 1.9, Fig 3A–C, Appendix Fig S12), then a branch point separating the remaining cells into differentiated cell types (score = 2.7). We also analysed the data obtained by Velten *et al* (2017), comprising a smaller number of cells (1,034 relative to 2,730 for the Paul *et al* dataset). TreeTop suggested a single branch point separating differentiated cell types into six branches (score = 1.5, Fig 3D–F, Appendix Fig S13). Here, all differentiated cell types are separated into individual branches, with the exception of Erythroid, Megakaryocyte and Eosinophil/Basophil cells, which comprise branch 1. The small branch cell counts (37–380 cells per branch) precluded recursive application of TreeTop. Taken together, these results are consistent with the findings of Paul *et al*, and also with additional work based on single cell lineage tracing, which indicates that the divergence between myeloid and erythroid cells begins within MPPs (Perié *et al*, 2015).

TreeTop is able to discriminate between processes with and without branch points, from datasets comprising up to millions of cell events, and identify multifurcations as well as multiple levels of branch points. We showed that alternative methods for branch analysis of single cell data can report branches, where they are known not to exist. By comparison with branching scores from non-branching synthetic data, TreeTop reduces the potential for unnecessary investigation of false-positive branching results. TreeTop provides support for the shallower hierarchy model of hematopoiesis, rather than the classical deep hierarchy based on oligopotent progenitors. The complexity of modern single cell data requires dedicated computational approaches such as TreeTop to identify possibly branched transitions between new cell types, and to reexamine assumed state trajectories in the light of high-dimensional technologies such as single cell sequencing and mass cytometry. Such approaches have led to reevaluations of long-standing previous hypotheses about the complexity and plasticity of cell types in both health and disease, including identification of new cell types such as emergency NK cells (Ohs *et al*, 2016) and human innate lymphoid subsets (Simoni *et al*, 2017), and transitions between them, such as myogenic progenitors (Porpiglia *et al*, 2017). TreeTop provides a tool based on a novel conceptual approach to assess the presence of branch points for biological processes observed via high-dimensional single cell datasets, and examine the findings via an interpretable layout.

We provide a MATLAB package for TreeTop (https://github.com/wmacnair/TreeTop).

# Materials and Methods

### Reagents and Tools table

| Reagent/Resource | Reference or Source | Identifier or Catalog Number |
|---|---|---|
| **Software** | | |
| MATLAB 2017a | MATLAB and Statistics Toolbox Release 2017a, The MathWorks, Inc., Natick, Massachusetts, United States | |
| R version 3.5.1 (2018-07-02) | R Core Team (2018) R: A language and environment for statistical computing. Vienna, Austria: R Foundation for Statistical Computing | |
| Monocle 2.10.0 | Reversed graph embedding resolves complex single cell trajectories, Qiu *et al*, *Nature Methods*, 2017 | |
| Wishbone 0.4.2 | Wishbone identifies bifurcating developmental trajectories from single cell data, Setty *et al*, *Nature Biotechnology*, 2016 | |

## Methods and Protocols

### TreeTop overview

This section describes the geometrical intuition motivating TreeTop. Intuitively, branch points can be thought of as the location in some space where three or more distinct state trajectories meet. In our case, the space is the state space of cells, consisting of possible vectors of species abundances for individual cells. In mass cytometry, species correspond to proteins, and in single cell RNA-seq, species correspond to mRNA trancripts. To identify such points, we sample an ensemble of random trees representing possible transitions in the dataset, then score every point based on how consistently it partitions the remaining points into distinct branches.

Briefly, we first sample an ensemble of random trees defined over a set of reference nodes, i.e. a representative subset of cell measurements. Each tree uses the same set of reference nodes; however, the connections between them may differ. In regions where data have consistent structure, many connections will be common across trees; in diffuse regions, we will observe high variability in connections. The ensemble of trees therefore captures how the subpopulations of cells may be connected to each other.

We use the learned ensemble of trees to look for branch points by considering each reference node in turn. For a given tree, we cut the tree at this reference node, partitioning the tree into branches. We then compare these induced branches across all trees, looking for consistency between them: a point where branching is observed will partition the remaining points into at least three branches which are similar across many trees, while one with no branching will show little similarity between the branches, or less than three induced branches. Hierarchical clustering on the consistency matrix allows us to identify these branches, and to quantify how consistent they are via a raw branching score.

Raw branching scores alone are not sufficient to assess whether a dataset is sampled from a branching process; they must be contextualized with information on typical scores from data for non-branching processes. To do this, we compare the scores of prospective branch points to scores derived from synthetic data with simple topologies not containing branch points. When the scores are the same or lower than those observed in the non-branching data, this suggests that no branching is present.

TreeTop fits trees to the data, which capture transitions between cell subpopulations. Important implicit assumptions of our method (and alternative methods) are therefore that the data are sampled from a continuous biological process, and that the data are sampled from the full range of the biological process of interest. If these criteria are not met, the cell subpopulations are separated in state space, and no evidence is available to determine the likely connections between them.

### Data preprocessing

Mass/flow cytometry data are arcsinh-transformed [with cofactors of 5 for mass cytometry (Bendall *et al*, 2014) and 150, or otherwise depending on the fluorescent tag, for flow cytometry]. Where the data are a mixture of non-overlapping components, the dimensionality reduction technique diffusion maps (Coifman *et al*, 2005) results in embeddings of the mixture components which are approximately orthogonal, with one diffusion component per mixture component (Schiebinger *et al*, 2015). This motivates preprocessing

data with diffusion maps, then taking the diffusion components with the largest eigenvalues as inputs to TreeTop.

There are many possible methods for reducing the dimensionality of single cell data. Given the wide range of processes from which they are sampled, we do not believe that is sensible to specify a universal recipe for upstream analysis. TreeTop is compatible with any selected preprocessing. We would advise trying multiple dimensionality reduction techniques to identify the one which best reflects prior biological knowledge about the data, and potentially also running TreeTop using each set of preprocessing options.

We have implemented several possible distance measures, including L1 (Manhattan distance), L2 (Euclidean) and angle distance. We have found little difference in results between L1 and L2; throughout this manuscript, we have used L1.

### Construction of ensemble of trees

TreeTop first selects reference nodes, which represent subpopulations of cells with particular profiles of molecular species expression. Initially, we perform density-based downsampling of the data, to remove outliers which could cause shortcuts, and to reduce bias towards species expression profiles more densely occupied by cells (as described in Qiu *et al* (2011)). The user must give an appropriate value of scale, $\sigma$, for calculation of density (TreeTop provides functionality to assist users in this decision). TreeTop then selects a small number $k$ of reference nodes, chosen to be evenly distributed through the data, thereby avoiding redundant concentrations of reference nodes in the same region. This is based on an algorithm developed for efficient initialization of $k$-means (Moseley *et al*, 2012).

We use $k = 200$ throughout this study, as a balance between too few nodes, which would not allow accurate representation of all subpopulations and transitions in the data, and too many, requiring extensive computation for little increase in resolution. We suggest using $k = 200$ where possible, and considering smaller values of $k$ for datasets with smaller numbers of cells. The number of nodes selected does not have a large influence on the branches identified, or the relative branching score (Appendix Fig S14). Many of the early single cell RNA-seq datasets comprise up to 200 cells. The construction of TreeTop consequently requires an even smaller number of reference cells, say 20. The problem proposed is then the identification of branches for a set of 20 nodes. This set is small and based on noisy and extremely high-dimensional data. If we reduced the dimensionality to 10 dimensions, for example, we are sampling 20 noisy points from a 10-dimensional space and asking whether they form connected branches. This is a too small number of datapoints for such a problem, and therefore, we do not recommend TreeTop analysis for small cell numbers (TreeTop is written to return an error when it is applied to data with < 1,000 cells). This problem is also largely resolved for the larger single cell RNA-seq datasets measured with current droplet-based technologies.

The remaining cells are then labelled according to their closest reference node, partitioning the dataset into a Voronoi tessellation. Density-downsampled cells are excluded for the purpose of selecting the $k$ nodes, but included for calculating the Voronoi tessellation; outliers are permanently excluded from the dataset.

TreeTop then samples a random ensemble of $n$ trees connecting these reference nodes. Within each Voronoi partition, one cell is selected uniformly at random, giving the same number of points as reference nodes, $k$. These are then joined by a minimum spanning

tree (MST), giving a unique set of edges which connect the reference nodes and do not contain cycles. For each tree, the edges identified are recorded in an adjacency matrix, where edge weights correspond to the distances between the selected cells.

### Ensemble of trees visualization

Visualization of the data is based on a force-directed graph layout algorithm. This class of approaches takes as input a graph, consisting of nodes connected by edges, where the edges may have weights associated with them. These can be viewed as springs, which are in a low energy state when the distance between the ends is similar to their weight, and in a higher energy state when it is different. Force-directed graph layout algorithms seek an embedding of the graph in a low-dimensional space that minimizes the resulting energy.

To apply this to TreeTop, we first take the mean over all edges in all trees in the ensemble, resulting in a "union" graph that has an edge between two nodes where that edge occurred in at least one of the trees. Each edge has two values associated with it: the proportion of trees in which it occurred, and the mean distance between the nodes across those edges. To improve clarity of the graph, we then remove edges that occurred with low frequency, applying the maximum possible frequency threshold that still results in a connected graph. We then apply a force-directed graph layout algorithm to this graph (Harel & Koren, 2001).

The union graph is effectively a superposition of all the trees in the ensemble. It typically is not a tree and can therefore contain cycles. To illustrate this point, consider a dataset sampled from the circumference of a circle (as in the third column of Appendix Fig S1). Here, Monocle fits a tree with no branches, but with a cutpoint at some point around the circle (Appendix Fig S1C). Each of the individual trees sampled by TreeTop also must include a cutpoint, at different points around the circumference, but taken together the topology they identify is correct (Appendix Fig S1D).

### Identification of branch points

We identify branch points by evaluating how consistently a given node partitions the other nodes into three or more branches, across all members of the ensemble of trees. We quantify this consistency by a branching score defined as follows. If we "cut" a tree by removing one of its nodes, by definition of a tree this partitions the remaining nodes into disconnected components, or "branches". Taking a node $x$ and removing $x$ in all $n$ trees $T_1,\ldots,T_n$ in the ensemble, we obtain $n$ partitions of the remaining nodes into induced branches. For all pairs of nodes $i, j \neq x$, we then calculate the proportion $B_{xij}$ of the trees in which $i$ and $j$ were assigned to the same branch. If $B_{xij}$ is close to 1, then cutting at $x$ often placed $i$ and $j$ into the same branch; if it is close to 0, then $i$ and $j$ were rarely placed into the same branch. If $x$ is a branch point, we should be able to cluster the points such that pairs of points taken from within a cluster have a high probability of being placed into the same branch, and pairs of points taken from different clusters have a low probability of being placed into the same branch. We quantify the extent to which each node $x$ satisfies this criterion via single-linkage hierarchical clustering, using the matrix $B_x$ as a similarity measure, which we use to calculate a raw branching score.

Our raw branching score is the mean size of the clusters indicating the presence of a branch point, namely the third largest (and any smaller) clusters; hereafter *branch clusters*. For a given set of similarity values, the threshold at which the dendrogram is cut determines the outputs of hierarchical clustering. The raw branching score for node $x$ is the mean size of the branch clusters across 100 cut thresholds over the interval [0, 1]; this assesses the average size of the branch clusters at a putative branch point. The node with the largest branching score is the identified branch node (and will correspond to the multiple individual cells assigned to that node).

Note that we evaluate the mean size of the third largest *and smaller* clusters, rather than just the size of the third largest cluster. This is to permit identification of multifurcations. Where a dataset is sampled from a process joining more than 3 branches, the size of the third branch is smaller: scoring on the basis of only the third largest cluster would make multifurcations more difficult to identify.

### Relative branching scores

We compare the raw branching scores of prospective branch points to distributions of scores derived from synthetic data with simple topologies not containing branch points, termed reference score distributions. By normalizing raw branching scores by the highest scores observed in comparable non-branching data, we obtain a relative branching score indicating the strength of evidence for branching in a given dataset.

#### Choice of synthetic reference data topologies for reference score distributions

The definition of reference score distributions must account for situations where data have no branch points but may have other structure. We initially tested permutations of the data [permutations of data are widely used in statistics to identify the size of an effect above background rates (Good, 1994)]. However, in this case permutations of the original data (by randomly permuting the set of values for each input dimension) also disrupted commonly present non-branching structures. In almost all real and synthetic data that we tested, this yielded extremely low scores that were lower than those observed in synthetic and real data known not to contain branch points (Appendix Fig S15). We addressed this problem by specifying reference score distributions from synthetic datasets with simple topologies without branch points, which would then result in the procedure rejecting the widest range of non-branching datasets.

We assume that the data are sampled from a continuous process, and that measurement of the data is noisy. To identify a reference score distribution which gives the highest possible raw branching scores, while not containing branch points, we therefore considered synthetic datasets sampled from relatively simple, connected non-branching topologies. As possible sets of reference score distributions, we considered embeddings of simple, non-branching, low-dimensional manifolds in a higher-dimensional space, with Gaussian noise. Analysis of the effect of dimensionality on the raw branching score shows that the highest raw branching scores derive from input datasets which are 2-dimensional manifolds embedded in a higher-dimensional space (Appendix Fig S16). Within 2-dimensional manifolds, we consider only convex manifolds, to exclude branching processes. Of these, we found that triangular input data gave the highest raw branching scores (details for generating this synthetic data are given in Appendix Table S1) and found that the score distributions from 2-dimensional triangular data consistently gave the highest branching scores (Appendix Figs S15 and S16). We therefore used reference score distributions based on triangular synthetic data for calculating the relative branching scores.

While we consider synthetic datasets which cover a wide range of simple non-branching topologies (0-, 1- and 2-dimensional non-branching manifolds with measurement noise added), we cannot exclude that more intricate non-branching topologies could also result in high raw branching scores. However, we expect the data actually observed are unlikely to follow such non-branching topologies, as any fine structures will be obscured by measurement and biological noise, which are known to account for much of the variation in single cell data (Elowitz et al, 2002).

*Defining a reference score distribution specific to the input dataset*
For a given input dataset, it is important that the reference score distribution used for calculating the relative branching scores is appropriate, i.e., it is based on synthetic data which has similar dataset-specific parameters to the input dataset. We consider the following relevant dataset-specific parameters:

1   number of single cell measurements ($n\_obs$),
2   dimensionality of the single cell measurements ($d$) and
3   standard deviation of the noise ($\rho$), comprising biological variation independent of the branching process and measurement noise.

In addition, we consider an analysis-specific parameter, namely the number of reference nodes for the TreeTop runs ($k$), to derive the final reference score distribution for calculation of relative branching scores.

This procedure requires identifying comparable dataset-specific parameters for a given input dataset. This is trivial for number and dimensionality of single cell measurements. However, accurately specifying the extent of noise (i.e. unwanted variation due to measurement and biological variability) is typically not feasible. We address this difficulty by choosing a reference score distribution to have the highest scores possible amongst all noise parameter values. We found that score distributions of raw branching scores derived from non-branching synthetic data without measurement noise have higher scores than those from datasets with non-zero measurement noise (Appendix Fig S17). This makes intuitive sense: the noise we add means that the synthetic data we generate is a mixture of an underlying manifold and a multivariate Gaussian, and we have observed extremely low scores for Gaussian-distributed synthetic data (Appendix Fig S15). If we knew confidently the type and degree of noise for the relevant data type (e.g. mass cytometry or single cell RNA-sequencing data), we could construct data type-specific reference score distributions, based on appropriately noisy synthetic data, which might then avoid some possible false negatives currently resulting from our conservative approach.

On this basis, we define an input dataset-specific reference score distribution for testing a new input dataset by using synthetic data with (i) zero measurement noise, and (ii) comparable settings for the remaining dataset-specific parameters, i.e. the number and dimensionality of single cell measurements. This dataset-specific reference score distribution is equally or more conservative than the ideal, dataset-specific reference score distribution that we would choose knowing the correct, but unavailable, measurement noise parameter.

*Calculation of relative branching scores*
For efficient processing of a new input dataset, we use a set of precomputed reference score distributions corresponding to a grid of dataset-specific parameter values covering the range we expect to see in new input datasets. For each individual input dataset, we choose the reference score distribution with the most similar parameter values to the input dataset. Specifically, for each possible combination of the following parameter lists, we applied TreeTop to 1000 randomly generated synthetic datasets with this parameter combination:

1   $n\_obs$ = 10,000, 20,000, 30,000, 40,000, 50,000, 60,000, 70,000, 80,000, 90,000, 100,000
2   $d$ = 5, 10, 15, 20, 25, 30
3   $k$ = 50, 100, 150, 200
4   $\rho$ = 0

For each TreeTop run, we took the maximum raw branching score observed, giving a score distribution of 1,000 maximum raw branching score values for each of these 240 combinations of parameter values (i.e. a set of 240 reference score distributions). For a given input dataset, we apply TreeTop to obtain the raw branching scores. We then take the maximum raw branching score observed for the input dataset $b_{max}$, and compare it to $b_{ref}$, the 95[th] percentile of the observed scores in the reference score distribution with the most similar parameter values, as a robust measure of the typical highest score in the reference datasets. We then report $b_{max}/b_{ref}$ as the relative branching score for this dataset.

We include this set of reference score distributions in our TreeTop package, and we have calculated a lookup table based on the range of numbers of observations, reference nodes and dimensionalities; for a given run, the package automatically selects the closest most conservative score distribution for comparison. The lookup table means that the test is not exactly specific to the data in question. However, we find the differences between score distributions that are neighbours in the lookup table to be small, and using a lookup table makes such an approach practical. The output from TreeTop, specifically the resulting relative branching score, is in principle conditional on the selected value of $k$. However, analysis of multiple runs on the same data with differing values of $k$ suggests that the results are not sensitive to this choice (Appendix Fig S14).

The range of dimensionalities made available in the lookup table is based on what is typical for flow and mass cytometry. Single cell RNA-seq data typically have much higher dimensionality, with several thousand features. However, techniques for assessing the overall structure of single cell RNA-seq datasets (rather than differential expression techniques, which seek to identify individual genes) have the starting assumption that the data exist on some lower-dimensional manifold. If much of the variance in the data cannot be explained by a lower-dimensional manifold, then the proposed macro-structure (such as trajectories) cannot be reliably inferred from the data. To run TreeTop on single cell RNA-seq datasets, we therefore first transform the data into a lower-dimensional representation, within the range of dimensionalities specified above.

*User interpretation of branching score outputs*
In Appendix Fig S3, we applied TreeTop to B-cell maturation data, based on PCA components of this data (first 10 PCs, accounting for 90% of variation), using TreeTop's default L1 distance. This resulted in a relative branching score of 0.97, which is just under the threshold of 1 and could be interpreted as weak evidence in favour of branching; this is consistent with the underlying biology. The graphical outputs of TreeTop show clear up-regulation of kappa and lambda chain expression along a linear trajectory. A larger

dataset that included more mature B cells would most likely show completely separated kappa and lambda clusters, and lead to a conclusion of branching from TreeTop.

TreeTop is intended to have a measure of conservativeness, in contrast to the other methods discussed. Although this may possibly result in under-reporting of branching (in this B cell example, the underlying biology is known to branch at some point in the process), it can be combined with user interpretation to suggest weak evidence of branching (i.e. where the relative branching scores are close to 1), which may indicate an area for further study.

### Multi-layer branch point identification

Recursive application of TreeTop can reconstruct deep hierarchies of branch points. TreeTop applied to a branching dataset identifies both a branch point, and the corresponding branches induced at this point. We can recursively apply TreeTop to each of the branches, potentially resulting in further branch points and branches within these (Fig 2B). For visualization, we use the force-directed graph layout for the whole dataset (i.e. the top-level application of TreeTop) and display all identified branch points obtained via recursive TreeTop application, with leaf nodes to show branches containing no further branching.

### TreeTop Pseudocode

1  Preprocessing of data
2  Calculate density of points, based on an appropriate $\sigma$
3  Density-based downsampling, removal of outlier cell events (as described in the section *Density-dependent down-sampling* within the Methods of Qiu *et al*, 2011)
4  Pick reference nodes via k-means + + (Moseley *et al*, 2012)
5  Voronoi partition of cells according to closest reference node
6  For $n$ from 1 to $N$, sample tree $T_n$:
   a  Uniformly at random pick one point from each Voronoi partition
   b  Join these by MST
   c  Record details of MST: adjacency matrix with distances, IDs of cells picked, any gating of selected cells
7  Generate force-directed graph layout embedding based on ensemble of trees
8  For every candidate branch point $x$:
   a  For every tree $n$:
      i  Cut at this branch point, giving branches $b(i) = b(x, n, i)$ for each point $i \neq x$
      ii  Record induced branches as matrix $B_{xn}$
         $(B_{xn})_{ij} = \delta_{b(i)b(j)}$
         (i.e. 1 where in same branch, 0 where different)
   b  Take mean of $b$s across all trees $n$, giving matrix
      $B_x = B_{xij} = P(i, j \text{ in same branch} \mid \text{cut at } x)$
   c  Do single-linkage hierarchical clustering using $B_x$ as similarity matrix, to generate dendrogram $D_x$.
   d  For each threshold $p_{cut} = 0.01, 0.02, \ldots, 0.99$
      i  Cut $D_x$ at this value.[1] Cutting at a value $p_{cut}$ induces a clustering of the points. When points $i,j$ are in different clusters, they have probability $< p_{cut}$ of being connected, on average over all trees.

   ii  Calculate the sizes of all induced clusters (i.e. the number of reference nodes in each cluster).
   e  Placed in descending order, these are $N_1(p_{cut}) \geq N_2(p_{cut}) \geq N_3(p_{cut}) \geq \ldots$. The raw branching score is defined as mean third largest or smaller branch size over all cut thresholds: $\frac{1}{99}\sum_{p_{cut}=0.01}^{0.99} \sum_{s \geq 3} N_s(p_{cut})$.
9  Compare number of observations and number of reference nodes to reference score distribution lookup table, to obtain closest most conservative reference score distribution.
10  Calculate relative branching scores by dividing raw branching scores for the input dataset by the 95[th] percentile of raw branching scores in the reference score distribution; if $> 1$, the reported branch point is the point with the highest relative branching score. The reported branches are those resulting from the threshold $p_{cut}$ which gave the largest third branch.

### Generation of hierarchically branching synthetic data

As a synthetic test case with known ground truth, we simulated expression data for proteins organized in a tree of binary toggle-switches. Each switch stochastically and mutually exclusively commits to expressing one of two proteins, which subsequently activates its downstream switch and branch, respectively. Therefore, one simulated trajectory mimics the multi-step differentiation process of one single cell. The structure and parameters of the underlying biochemical model were adapted from (Ocone *et al*, 2015; supplementary section 2.2.1).

Each protein is modelled with basal production, Hill-type functions for activation (from upstream) and inhibition (for switch) and mass-action degradation. A protein is up-regulated if activated from upstream and not inhibited within the switch:

$$\alpha \; \frac{g_U(t)^{h_+}}{g_U(t)^{h_+} + \kappa_+^{h_+}} \frac{\kappa_-^{h_-}}{g_s(t)^{h_-} + \kappa_-^{h_-}},$$

where $\alpha$ is the basal production rate, $g_U$ is the upstream protein, $g_S$ is the other protein in the switch, and $\kappa$ and $h$ are the dissociation constants and Hill coefficients of the activation (+) and inhibition (−) according to Appendix Table S2.

The simulation was performed with tau-leaping (Gillespie, 2001), an approximate stochastic simulation algorithm, implemented in matLeap (preprint: Feigelman *et al*, 2016). We simulated 100,000 trajectories with initial counts for each protein drawn from the Poisson distribution ($\lambda = 100$). The protein abundances were saved at 100 uniformly placed time points from $t = 0$ to $t = 150$.

### Analysis of specific datasets

Parameter settings for each of the runs presented in this paper are described in Appendix Table S3 for TreeTop, and Appendix Table S4 for Wishbone.

For Monocle, all data were pre-processed via arcsinh transform, as for TreeTop. For each dataset, a sample of 2,000 cells was taken uniformly at random (Monocle becomes slow for large datasets, making downsampling necessary). Monocle was run using the Gaussian family, and with default values for other parameters. For the plots in this paper, the branch point which maximized the size

---

[1]Here we mean cutting in the sense used regarding hierarchical clustering, and not in the sense previously used for trees.

of the smallest branch was chosen manually for each Monocle output. Monocle was first published in 2014 (Trapnell *et al*, 2014), but has since been updated to Monocle 2 (Qiu *et al*, 2017; our analysis used Monocle 2, which for brevity we have referred to throughout the manuscript as Monocle).

We performed a comparison of algorithm timings, using the mass cytometry bone marrow data as an illustrative example (Appendix Table S5). We ran each method 10 times with different random seeds. TreeTop and Wishbone were both applied to the full dataset, and complete in comparable lengths of time; Monocle 2 requires less time, but is applied to a severely downsampled dataset (2,000 cells relative to 100,000 cells). Times for the other datasets are similar. We note that we cannot assess the other methods in terms of how well they detect the presence or absence of branch points, as the other methods were not designed for this purpose. The comparisons shown are included to show a practical aspect of using TreeTop, and not intended to be an assessment of alternative methods applied to the same task.

## Data availability

A MATLAB package for TreeTop is available on GitHub: (https://github.com/wmacnair/TreeTop).

Expanded View for this article is available online.

## Acknowledgements

Will Macnair and Laura de Vargas Roditi were supported by the SystemsX.ch RTD PhosphonetPPM grant. Laura de Vargas Roditi was further supported by the ETH fellowship FEL-32 14-1 (ETH foundation).

## Author contributions

WM developed the method, performed the analysis and wrote the paper and the supplement. LDVR helped develop the tree ensemble sampling method, and gave feedback on the paper. SG simulated the synthetic branching data and wrote the relevant methodology section, and gave feedback on the paper. MC supervised the study, contributed to the method development and wrote the paper.

## Conflict of interest

The authors declare that they have no conflict of interest.

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
