## [Review Process File · Molecular Systems Biology]

Tree-ensemble analysis assesses presence of multifurcations in single cell data

Will Macnair, Laura De Vargas Roditi, Stefan Ganschä and Manfred Claassen

Review timeline:

Submission date:	12 th July 2018
Editorial Decision:	24 th September 2018
Revision received:	16 th January 2019
Editorial Decision:	22 nd February 2019
Revision received:	26 th February 2019
Accepted:	26 th February 2019

Editor: Maria Polychronidou

Transaction Report:

1st Editorial Decision

24th September 2018

Thank you again for submitting your work to Molecular Systems Biology. We have now heard back from two of the three referees who agreed to evaluate your study. Unfortunately, after a series of reminders we did not manage to obtain a report from reviewer #1. In the interest of time, and since the recommendations of reviewers #2 and #3 are quite similar, we have decided to proceed with these two reports. As you will see below, the reviewers acknowledge that the presented approach seems potentially useful for the field. They raise however a series of concerns, which we would ask you to address in a major revision.

I think that the recommendations of the reviewers are rather clear so there is no need to repeat the points listed below. All issues raised by the reviewers need to be satisfactorily addressed. As you may already know, our editorial policy allows in principle a single round of major revision so it is essential to provide responses to the reviewers' comments that are as complete as possible. Please feel free to contact me in case you would like to discuss in further detail any of the issues raised by the reviewers.

REFeree REPORTS

Reviewer #2:

The authors present an algorithm for finding developmental branching points in high dimensional single cell data such as mass cytometry or scRNAseq. The main shortcoming in previous approaches that they see their algorithm as addressing is the lack of a method of assessing the validity of branch points. They also are concerned with the necessity of choosing a root or start point in some

algorithms

(Wishbone) and the "strong topological assumptions" of other algorithms such as Monocle 2.

The basic steps of the algorithm:

1. Data is transformed and dimensionally reduced with diffusion maps.
2. Density dependent down sampling is carried out as in SPADE.
3. Reference nodes (cells) are chosen so as to be evenly distributed in the cloud of cells.
4. The cells are partitioned by assigning each to the closest reference node.
5. Generate ensemble of trees. To generate one tree, randomly pick one cell from each partition and construct the minimum spanning tree (MST) using distances between chosen cells.
6. Data for the scoring of potential branch points are then gathered cutting each tree in the ensemble at each tree node and recording how many times each pair of cells ends up in the same branch for each node. This data is summarized in a matrix B_{xij} , with x being a node and i and j a pair of cells.
7. A score is then calculated for each node based on the matrix B_{xij} . The meaning of the scores is then assessed by comparison to scores for nodes in synthetic datasets without branch points. On this basis it is decided if the highest scoring node is a valid branch point.
8. Given that a branch point is found, a search is made for other branch points by applying TreeTop recursively to the branches emanating from the found branch point.

The innovative portions of this algorithm are steps 5 through 8 where the main contribution is the automatic scoring of potential branch points to avoid false positives. TreeTop seems to perform well on hierarchically branched synthetic data. Further to this, the authors tested Treetop on several previously published data sets: T cell maturation in the thymus from the Wishbone paper (reference 15), B cell maturation data from the Wanderlust paper (reference 9) and healthy bone marrow data from the viSNE paper (reference 22). While some of the results were mixed for these datasets based on expected biology (see major comments) for the T cell maturation data TreeTop recovers the expected branching to CD8 and CD4 as found by Wishbone.

Overall, TreeTop is a potentially compelling new algorithm that can quantitatively identify multiple branch points in high dimensional single cell datasets, however more evidence is needed to show that TreeTop is able to accurately and sensitively assess their validity. The manuscript would be significantly improved if its performance in this respect could be quantitatively compared to already existing data visualizations that have similar capabilities.

Major comments

- TreeTop finds no branch point in the B cell maturation data. This is problematic as a 3D PCA (first three PCA components) plot of this data with cells colored either for Kappa or Lambda clearly shows obvious well-separated Kappa and Lambda branches. Moreover, the bone marrow data in figure 2 seems underbranched compared to previous representations of that dataset and the expected biology. Together, these suggest at a minimum that the TreeTop algorithm is overly conservative in assessing branch points and that false-negatives could be an issue.

- Presumably, the generation of an ensemble of trees is what the authors mean by avoiding strong topological assumptions as in Monocle, which just generates one tree. On the other hand, the authors at one point state that the "embeddings found by Monocle identify exclusively trees, regardless of the topology of the data". This is confusing because TreeTop also only identifies trees, although an ensemble of them. Please clarify and elaborate.

- The synthetic data sets here are constructed to match numbers of cells, dimensionality of the single cell data, and standard deviation of the noise in the data. The synthetic data is constructed not to have branch points and various underlying dimensionalities: 0,1,2. Given that branch point identification is a key contribution, it is not clear that the synthetic data provides an adequate reference to assess the meaning of the scores for the actual data set of interest.

Minor comments

- The TreeTop force directed data visualization does not seem very clear in comparison to, for example, The Gephi forced directed visualization used in X-Shift.

Reviewer #3:

In this paper, Macnair et al. propose a new method TreeTop to identify and quantify branching point of biological processes from single-cell mass cytometry and RNA sequencing data. In addition, TreeTop also provides a graph-based method to visualize the learned ensemble of trees. TreeTop is able to overcome the limitations of existing approaches, including 1) the inability to detect the presence of a branch point; 2) the strong topological assumption 3) the supervised root point selection. The authors have tested TreeTop on previously published datasets depicting different biological processes including T cell maturation, B cell differentiation and hematopoiesis, as well as on synthetic datasets of different topologies proving their method's superiority and robustness. The authors also compared the performance of TreeTop to other two methods including wishbone and monocle for assessing the presence of branch points and global structure. In general, it is potentially useful in avoiding false positive branch points for many single-cell trajectory inference studies.

Major comments:

1. The methodology of branch identification in TreeTop, which mainly consists of density-based downsampling, building MST, constructing consistency matrix and deciding the final clusters, shares some similarities with the method Éclair (Giecold, G., Marco, E., Garcia, S.P., Trippa, L. & Yuan, G.C. Robust lineage reconstruction from high-dimensional single-cell data. *Nucleic Acids Res* (2016).) It would be worthwhile to compare it to Éclair and prove TreeTop's advantages over Éclair.
2. The authors agreed on the popularity of single-cell RNA-sequencing in studying single-cell transcriptional profiles. But in this paper, TreeTop is only tested on one RNA-seq dataset. Given the prevalence of scRNA-seq, I would recommend that the authors add more scRNA-seq analyses to make the experimental results more convincing.
3. In the preprocessing step, for single cell mass cytometry data, top diffusion components are used for T cell thymic maturation but not for the others. It would be helpful if the authors can explain how they decided whether to use diffusion map in the preprocessing steps.
4. Page 11, Paragraph 1. Last sentence 'The node with the largest branching score is the identified branch point.' To my understanding, each 'node' should contain many different 'points(cells)' within the corresponding Voronoi partition. Then which 'point/cell' should be used as the branch point? Or does the 'node' here have different meaning from the 'reference node'?
5. TreeTop needs to use a set of precomputed reference score distribution, which are dataset-specific and based on triangular synthetic data, to process new input data. This sounds more empirical and lacks solid proof. Given some more complex non-linear and concave non-branching structure (e.g. swiss roll), will it still work?
6. For the multi-layer branch point identification, instead of recursive application, is it possible to only run TreeTop once to get the multi-layer hierarchy based on the same branching score threshold? If not, what's the advantage of recursive application? Is it true that for the same point its branching score tends to get higher with the recursive division of initial tree? If that's the case, is it still fair to compare them directly after reassembling the subbranches since they are calculated in different configurations?
7. For TreeTop package, I ran it without success in matlab 2014b and got the following errors after strictly following the authors' tutorial. Hope the authors could solve this issue in their potential new version.

```
>> version
```

```
ans =
```

```
8.4.0.150421 (R2014b)
```

```
>> treetop_pre_run(input_struct, options_struct)
```

```
running pre-run analysis for TreeTop
```

```
1/6 Getting data
```

```
opening 1 files:
```

```
.
```

```
combining into one matrix
```

```
2/6 Plotting marginals of used markersUndefined function 'plot_fig' for input arguments of type
```

```
'matlab.ui.Figure'.
```

```
Error in treetop_pre_run>plot_marginals (line 41)
```

```
plot_fig(fig, plot_stem, options_struct.file_ext, fig_size);
```

Error in treetop_pre_run (line 16)
 plot_marginals(all_struct, input_struct, options_struct)

Minor comments:

- (1) The notation k is inconsistent in this paper. Page 10, k is the number of nodes. Page 14, k is the tree number. This can cause many confusions.
- (2) The authors should mention the reason why monocle is only applied to the sample of 2000 cells instead of the full dataset.
- (3) Page 9, the last sentence in last paragraph 'This is problem is also largely resolved for the larger single cell RNAseq datasets measured with current droplet-based technologies.'. It's grammatically wrong.

1st Revision - authors' response

16th January 2019

Response summary to the referees' comments for manuscript MSB-18-8552

This is the response to the reviewer comments for the paper entitled "Tree-ensemble analysis assesses presence of multifurcations in single cell data".

We thank the reviewers for their overall positive evaluation of the novelty and significance of our contribution, as well as for their constructive suggestions. We have identified and addressed the following main issues raised by the reviewers:

1. **The considered synthetic reference data may not be an adequate reference for assessment of bifurcation presence in real data, and may result in overly conservative branch point identification.**

Branching processes in single-cell data are complex patterns. Many statistical significance tests use permutations of the experimental data, however we found that assessing the presence of branch points with permutations results in over-reporting of branch points. We therefore designed synthetic reference data for branch point assessment which includes non-branching structures. We demonstrate that branch point assessment based on our approach leads to correct identification of branch point presence as well as absence for a range of examples of branching and non-branching processes.

TreeTop was specifically designed to show some conservativity, as a benefit to users, in contrast to algorithms such as Wishbone (which always report branches). We sought to reduce spurious reports of branching which would lead to wasted time and resources for additional validation experiments. We have demonstrated that TreeTop is able to identify known branch points in multiple datasets, and therefore believe that TreeTop is appropriately, rather than overly, conservative.

2. **TreeTop should be demonstrated on more single cell RNA-seq datasets.**

We agree with the reviewer that given the ever-increasing use of single cell RNA-seq technologies, a more comprehensive demonstration of TreeTop on this type of data would be beneficial. We have therefore revised the manuscript to include analysis of an additional dataset sampled from hematopoiesis (see new Figure 3 in revised manuscript).

The individual comments are discussed in turn below.

Reviewer 2

- 2.1 "TreeTop finds no branch point in the B cell maturation data. This is problematic as a 3D PCA (first three PCA components) plot of this data with cells colored either for Kappa or Lambda clearly shows obvious well-separated Kappa and Lambda branches."

Reviewer 2 is concerned that our analysis of maturing B cells (Supp Fig 3, original manuscript) may be overly conservative. Maturing B cells are known to express either kappa or lambda light chains. Reviewer 2 argues that these separate fates are clearly distinguishable in the dataset in question, however we disagree with the reviewer on this conclusion. In addition, the results from TreeTop

applied to this dataset suggest weak evidence of branching, which is entirely consistent with biological expectations of this dataset.

In applying TreeTop to the B cell maturation data, we followed the analysis in the original publication, which evaluated dissimilarity of cells with cosine distance (rather than Euclidean or L1 distance). Following the reviewer's suggestion, we have plotted the first 3 PCA components for this data (see below). We observe clear *clusters* corresponding to high Kappa and high Lambda cells, but do not agree that these form clearly separate *branches*; without prior biological expectations, this is not a clear conclusion from the PCA plot.

However, we have updated the application of TreeTop in the manuscript to be based on the first 10 PCs (as these account for 90% of explained variance) of the B cell data using L1 distance (TreeTop's default). TreeTop identifies the kappa/lambda clusters noted by the reviewer, however the confidence score for branching in the dataset is just below the cutoff used by TreeTop (updated Appendix Figure S3, p27 in revised manuscript). We interpret this as weak evidence in favour of a branching process, a conclusion which might change if the data was sampled from the full B cell differentiation process (i.e. with greater sampling of the Kappa/Lambda-committed cells).

This example demonstrates the utility of one of the outputs from TreeTop, the branching score. The default score threshold would report the evidence for the bifurcation of the kappa/lambda clusters as just below the threshold for branching; the user could then identify this borderline situation and follow up if applicable.

2.2 “Moreover, the bone marrow data in figure 2 seems underbranched compared to previous representations of that dataset and the expected biology.”

The bone marrow data in TreeTop Figure 2 is taken from the healthy control sample shown in Figure 6d of Amir *et al.* (Amir *et al.* 2013). The cell types identified in this data were: progenitor cells, T cells, CD20+ B cells, CD20- B cells, monocytes, NK cells and ungated cells. This dataset does not include markers which allow CD4+ and CD8+ T cells to be distinguished, which results in one fewer branch point. Our analysis does not clearly separate CD20+ and CD20- B cells, but these cells are also not clearly separated in the original paper. Our analysis is therefore consistent with previous representations of this dataset.

2.3 “Together, these suggest at a minimum that the TreeTop algorithm is overly conservative in assessing branch points and that false-negatives could be an issue.”

TreeTop provides a score of confidence for identified branch points. This score is designed to be rather conservative than too optimistic, which assists users by reducing possible investigations of false positive branch detections. This feature is in contrast to currently available algorithms which always report branches, without any score of confidence, leaving the calling of bifurcations a subjective user decision. Additional experiments are costly, and therefore we sought to reduce spurious reports of branching which would lead to wasted time and resources. Even where TreeTop reports no evidence of branching, Wishbone reports multiple branches in all cases, and therefore

cannot be used for discovery of new branches (p3, para 2 and Appendix Figure S1 in revised manuscript). This demonstrates that Wishbone is insufficiently conservative. We also showed that TreeTop is able to identify known branch points in multiple datasets, demonstrating that TreeTop is not overly conservative in these cases. Our analysis above shows that there is only weak evidence of a branch point in the B cell data, as demonstrated by TreeTop. Taken together, these results show that TreeTop is appropriately, rather than overly, conservative.

2.4 “Presumably, the generation of an ensemble of trees is what the authors mean by avoiding strong topological assumptions as in Monocle, which just generates one tree. On the other hand, the authors at one point state that the “embeddings found by Monocle identify exclusively trees, regardless of the topology of the data”. This is confusing because TreeTop also only identifies trees, although an ensemble of them. Please clarify and elaborate.”

TreeTop produces both visualizations and a relative branching score, both based on the ensemble of trees.

For visualization, the ensemble of trees is summarized in a ‘union’ graph, which is effectively a superposition of all the trees in the ensemble (see Methods section *Ensemble of trees visualization*, p12 revised manuscript, for details on additional pruning of low-frequency edges). This union graph typically is not a tree and can contain cycles. To illustrate this point, consider a dataset sampled from the circumference of a circle (as in the third column of Appendix Figure S1). Here, Monocle fits a tree with no branches, but with a cutpoint at some point around the circle (Appendix Figure S1c). Each of the individual trees sampled by TreeTop also must include a cutpoint, at different points around the circumference, but taken together the topology they identify is correct (Appendix Figure S1d).

Scoring of nodes as potential branch points is based on analysis across the ensemble of trees (see section *Identification of branch points* in revised manuscript). Analysis across the ensemble, instead of a single tree alone, enables identification of consistent non-spurious branching structure. For a given node, each tree is cut at that node, separating that tree into branches. The branching score reflects large and consistent branches across the ensemble. In the case of the circle, the branches would not be consistent across the ensemble. Additionally, the branching score is based on the size of the third largest branch (as at least three branches are necessary to make a branch point). In this example, any third branches only result from noise in the data, resulting in low scores.

2.5 “The synthetic data sets here are constructed to match numbers of cells, dimensionality of the single cell data, and standard deviation of the noise in the data. The synthetic data is constructed not to have branch points and various underlying dimensionalities: 0,1,2. Given that branch point identification is a key contribution, it is not clear that the synthetic data provides an adequate reference to assess the meaning of the scores for the actual data set of interest.”

Stated simply, the task which TreeTop addresses is to decide which of the following statements is true: this dataset contains a branching point, or this dataset does not contain a branching point. In principle, making this decision requires knowing the distribution of *all possible* non-branching datasets. Defining this distribution is a fundamentally difficult, poorly-defined problem, even if we make reasonable assumptions regarding measurement noise (meaning that we could exclude extremely contrived, biologically unrealistic examples constructed purely to result in high branching scores), continuity of data, and biological plausibility.

Standard statistical approaches address the problem of an unknown null distribution by permutations of the input dataset. However, permutations of the input data are not appropriate for assessing the presence of branch points, as permutations result in a complete loss of structure in the data: in addition to loss of structure induced by branch points, simpler structure such as that resulting from a dynamical process is lost. A consequence of this is that using permutation-based testing, structures in non-branching datasets would be reported as branch points. We therefore conclude that permuted input data is an incomplete approximation of the distribution of non-branching datasets, leading to a high rate of false positive branch point discoveries (see revised manuscript Appendix Figure S15, and section *Choice of synthetic reference data topologies for reference score distributions*, p15 paragraph 2).

We therefore sought to derive synthetic reference datasets which were non-branching, but resulted in the highest possible scores, meaning that when applied to real data we would minimize the number of false positive branch points reported. We considered simple, connected non-branching topologies, namely embeddings of simple, non-branching, low-dimensional manifolds in higher-dimensional space, with Gaussian noise. We showed that increasing the dimensionality or increasing the number of points considered in the synthetic distribution results in lower scores (see revised manuscript Appendix Figure S16, p46). This informed the use of the selected reference topologies,

which therefore exclude the widest range of non-branching datasets. To ensure that the topologies were appropriate to compare to a given input dataset, we calculated scores for the defined topologies for synthetic datasets whose data parameters (e.g. number of cells, dimensionality, number of reference cells) matched the input data.

In summary, defining the distribution of all non-branching, biologically plausible datasets is a difficult, unsolved problem. We have sought to address this gap by defining and identifying high-dimensional datasets with non-branching structure, which result in the highest possible branching scores, therefore minimizing possible false positive results. For accurate branch point identification in a new input dataset, we require that branch scores exceed all scores observed in non-branching datasets. We acknowledge that this procedure makes the assumption that we have considered all possible non-branching datasets for comparison. We have made an extensive empirical effort towards fulfilling this assumption and have demonstrated correct identification of presence as well as absence of branch points in synthetic and mass cytometry datasets. Assessment of branch point presence is a new, difficult and so far unaddressed problem to solve, and we present here a viable solution towards resolving it.

2.6 “The TreeTop force directed data visualization does not seem very clear in comparison to, for example, the Gephi force-directed visualization used in X-Shift.”

X-Shift was developed specifically as a tool for visualization, and while TreeTop includes a visualization component, its primary focus is branch analysis. In principle, the Gephi force-directed visualization could be applied to the graph learned by TreeTop, or equally, the branching scores learned by TreeTop could be displayed over the k-nearest neighbours graph used by X-Shift. (The outputs from running TreeTop allow both of these possibilities in the following files: the scores for each node in *[RUN_LABEL] branching scores.txt*, the union graph learned in *[RUN_LABEL] freq_union_tree.mat*, and the locations of the reference cells in *[RUN_LABEL]_mean_used_markers.txt*.)

Reviewer 3

3.1 “The methodology of branch identification in TreeTop, which mainly consists of density-based downsampling, building MST, constructing consistency matrix and deciding the final clusters, shares some similarities with the method Éclair (Giecold et al., *Nucleic Acids Res* (2016).) It would be worthwhile to compare it to Éclair and prove TreeTop's advantages over Éclair.”

We downloaded Eclair and attempted to run it on some sample data. However, we were unable to get it to run successfully, and the lead author no longer works in academia. Researchers comparing trajectory analysis packages were also unable to successfully run Eclair ((Saelens et al. 2018), p2).

3.2 “The authors agreed on the popularity of single-cell RNA-sequencing in studying single-cell transcriptional profiles. But in this paper, TreeTop is only tested on one RNA-seq dataset. Given the prevalence of scRNA-seq, I would recommend that the authors add more scRNA-seq analyses to make the experimental results more convincing.”

We have applied TreeTop to the Paul *et al.* dataset (see new Figure 3, panels a-c in revised manuscript). This comprises single cell RNA-seq data from 2730 developing myeloid cells, labelled as follows: granulocyte-macrophage progenitor (GMP), megakaryocyte-erythroid progenitor (MEP), erythrocytes (Ery), dendritic cells (DC), monocytes (Mo), basophils (Baso), neutrophils (Neu), eosinophils (Eos), megakaryocytes (Mk) and lymphocytes (Lymph). Here, TreeTop finds a branch point (comprising primarily of MEPs) which separates erythrocytes, megakaryocytes and other cells, then a further branch point separating these cell types. This is consistent with findings by other authors, for example (Perié et al. 2015).

3.3 “In the preprocessing step, for single cell mass cytometry data, top diffusion components are used for T cell thymic maturation but not for the others. It would be helpful if the authors can explain how they decided whether to use diffusion map in the preprocessing steps.”

There are many possible methods for reducing the dimensionality of single cell data. Given the wide range of processes from which they are sampled, we do not believe that is sensible to specify a universal recipe for upstream analysis. TreeTop is compatible with any selected pre-processing. We would advise trying multiple dimensionality reduction techniques to identify the one which best reflects prior biological knowledge about the data, and potentially also running TreeTop using each set of pre-processing options. In this specific case, the cells corresponding to maturing CD4+ and CD8+ T cells were much more clearly separated in the diffusion map components, than in the PCA components.

- 3.4 Page 11, Paragraph 1. Last sentence 'The node with the largest branching score is the identified branch point.' To my understanding, each 'node' should contain many different 'points(cells)' within the corresponding Voronoi partition. Then which 'point/cell' should be used as the branch point? Or does the 'node' here have different meaning from the 'reference node'?

We thank the reviewer for identifying this lack of clarity. A node contains multiple cells within the Voronoi partition. Our phrasing here was insufficiently clear, and would have been better phrased as "... is the identified branch node"; this is amended in the revised manuscript. We believe identifying a group of cells as branch points makes the most biological sense (rather than a single cell). (p14, para 4 in revised manuscript)

- 3.5 "TreeTop needs to use a set of precomputed reference score distribution, which are dataset-specific and based on triangular synthetic data, to process new input data. This sounds more empirical and lacks solid proof. Given some more complex non-linear and concave non-branching structure (e.g. swiss roll), will it still work?"

TreeTop branch point analysis is based on triangular synthetic data, since this topology has shown to be the most confounding compared to other considered non-branching topologies.

Further, since TreeTop is based on a neighborhood graph structure (ensemble of trees), it is unaffected by characteristics of the dataset which do not affect the underlying topology (i.e. which do not change the neighbourhoods of cells). Non-linear, concave or other structures which do not have branching, still do not have branching topologies.

As suggested by the reviewer, we have applied TreeTop to a 10-dimensional swiss roll dataset as an empirical confirmation of this point. The plot below shows the results of applying TreeTop, annotated by the angle around the Swiss roll; here, TreeTop recapitulates the known topology and does not report any branching. This dataset is included in the example data included on the TreeTop GitHub page.

- 3.6 "For the multi-layer branch point identification, instead of recursive application, is it possible to only run TreeTop once to get the multi-layer hierarchy based on the same branching score threshold? If not, what's the advantage of recursive application? Is it true that for the same point its branching score tends to get higher with the recursive division of initial tree? If that's the case, is it still fair to compare them directly after reassembling the subbranches since they are calculated in different configurations?"

The branching score calculated by TreeTop is based on average sizes of any consistent branches at a given point. In a dataset with a hierarchy of branch points, branch points lower in the hierarchy will by definition have smaller branches associated with them. This means that one threshold cannot be used to detect all branches, although each branch point will be a local maximum of branching scores. The runs of TreeTops for different subbranches use comparison datasets with appropriate numbers of cells, making the scores comparable.

- 3.7 "For TreeTop package, I ran it without success in MATLAB 2014b and got the following errors after strictly following the authors' tutorial. Hope the authors could solve this issue in their potential new version."

We apologize for this, and thank the reviewer for supplying the error log. Our testing of the package was clearly not sufficient! This was an issue with required subfolders not being automatically on the path, and is now fixed.

3.8 “The notation k is inconsistent in this paper. Page 10, k is the number of nodes. Page 14, k is the tree number. This can cause many confusions.”

Thank you for spotting this typo, now corrected. (p18 of revised manuscript, section *TreeTop Pseudocode*)

3.9 “The authors should mention the reason why monocle is only applied to the sample of 2000 cells instead of the full dataset.”

Monocle becomes slow for larger datasets. We have included a note to this effect in the revised manuscript. (p19 para 5 in revised manuscript)

3.10 “Page 9, the last sentence in last paragraph 'This is problem is also largely resolved for the larger single cell RNAseq datasets measured with current droplet-based technologies.'. It's grammatically wrong.”

We thank the reviewer for spotting this typo, which is now corrected. (p3, para 2 in revised manuscript)

2nd Editorial Decision

22nd February 2019

Thank you for sending us your revised manuscript. We have now heard back from the two referees who were asked to evaluate your study. As you will see below, the reviewers are satisfied with the modifications made and they think that the study is now suitable for publication.

Before we formally accept your study for publication we would ask you to address the following minor issues.

REFeree REPORTS

Reviewer #2:

Based on the revision and response the authors give a good explanation of the issues involved in devising a methodology to determine the validity of potential branch points.

We acknowledge that the problem is very difficult and they have made a good worthwhile initial contribution to the solution of this problem. While the method may be somewhat conservative, since they do report scores of potential branch points, users may judge marginal cases for themselves. For instance, this is the case for the K / Λ branching in the B cell data, discussed in section 2.2, where they point out that their method does indicate the marginal possibility of a branch point. Their answers in 2.2 and 2.4 about branching and topologies are also reasonable.

Overall, the work in the paper presents a reasonable contribution to the analysis of branch points in single cell data and represents a unique contribution to the growing field of single cell trajectory analysis.

Reviewer #3:

In the revised manuscript, the authors have well addressed my concerns sufficiently. Great work. I would recommend it for publication.

Corresponding Author Name: Manfred Claassen

Manuscript Number: MSB-18-8552